# Spectral learning of Bernoulli linear dynamical systems models

**Iris Stone**[*]                                                                                     *istone@princeton.edu*
*Princeton Neuroscience Institute*
*Princeton University*

**Yotam Sagiv**[*]                                                                                    *ysagiv@princeton.edu*
*Princeton Neuroscience Institute*
*Princeton University*

**Il Memming Park**                                                     *memming.park@research.fchampalimaud.org*
*Champalimaud Foundation*

**Jonathan Pillow**                                                                                  *pillow@princeton.edu*
*Princeton Neuroscience Institute*
*Princeton University*

**Reviewed on OpenReview:** *https: // openreview. net/ forum? id= giw2vcAhiH&noteId=FNX7ncpglJ*

## Abstract

Latent linear dynamical systems with Bernoulli observations provide a powerful modeling framework for identifying the temporal dynamics underlying binary time series data, which arise in a variety of contexts such as binary decision-making and discrete stochastic processes (e.g., binned neural spike trains). Here we develop a spectral learning method for fast, efficient fitting of probit-Bernoulli latent linear dynamical system (LDS) models. Our approach extends traditional subspace identification methods to the Bernoulli setting via a transformation of the first and second sample moments. This results in a robust, fixed-cost estimator that avoids the hazards of local optima and the long computation time of iterative fitting procedures like the expectation-maximization (EM) algorithm. In regimes where data is limited or assumptions about the statistical structure of the data are not met, we demonstrate that the spectral estimate provides a good initialization for Laplace-EM fitting. Finally, we show that the estimator provides substantial benefits to real world settings by analyzing data from mice performing a sensory decision-making task.

## 1 Introduction

Latent linear dynamical system (LDS) models are an important and widely-used tool for characterizing the structure of many different types of time series data, including natural language sequences (Belanger & Kakade, 2015), task-guided exploration (Wagenmaker et al., 2021), and neural population activity (Gao et al., 2015; Nonnenmacher et al., 2017; Zoltowski et al., 2020). To fit these models, the typical approach is to use an inference method such as the expectation maximization (EM) algorithm in order to find a local maximum of the likelihood function (Baum et al., 1970; Dempster et al., 1977; Shumway & Stoffer, 1982). However, such inference methods are often time-consuming, computationally expensive, and provide no guarantee of finding the global optimum. To address these issues, an active area of research has emerged using spectral methods to identify the parameters of linear time invariant (LTI) systems from input-output data (Martens, 2010; Anandkumar et al., 2014; Belanger & Kakade, 2015; Hazan et al., 2017; 2018). Such methods rely

---

[*]Equal contribution

on an approach from control theory known as subspace identification (SSID), which uses moment-based estimation to fit the model parameters without the need for an iterative procedure (Ho & Kalman, 1966; van Overschee & de Moor, 1994; Viberg, 1995; van Overschee & De Moor, 1996; Qin, 2006). The resulting parameter estimates are consistent and avoid the problem of local optima that are common with inference methods such as EM, although they typically suffer from lower accuracy. Therefore, a common strategy is to combine both approaches, using the SSID estimates as initializations for EM in order to speed convergence and increase the likelihood of finding the global optimum.

Most recent developments in spectral estimation methods for LDS models have focused on undriven (no inputs) linear-Gaussian observation processes. However, there has been work extending SSID for LDS models to handle Poisson observations, which is a useful framework for modeling neural spike-train data at certain timescales (e.g. > 10ms), as well as developing the general case for input-driven models (Buesing et al., 2012). Yet there is a need for additional research explicitly expanding spectral methods to other distribution classes. For example, binary time series data are common in many fields, including reinforcement learning and decision-making. While there has been some work exploring second-order models for binary variables (Bethge & Berens, 2008; Macke et al., 2009; Schein et al., 2016), there have been few attempts to develop a spectral learning method for LDS models with Bernoulli observations. Such a method would be extremely useful for identifying the dynamics of any binary time series data, including sequences of choice behavior and neural spike-trains with small bin sizes. In addition, recovery accuracy of the input-related parameters has not been well-characterized for non-Gaussian emissions models.

Here, we achieve two main innovations. First, we extend the SSID method to the probit-Bernoulli LDS case, deriving a new estimator: bestLDS (BErnoulli SpecTral Linear Dynamical System). From a technical perspective, this involves overcoming difficulties due to redundancies in the moments that are not present in the Poisson or general cases. We show this method yields consistent estimates of the model parameters when fit to input-output data from a wide range of simulated datasets. Furthermore, we demonstrate that using the outputs from bestLDS as initializations for EM significantly accelerates convergence. Second, we present new analyses, looking at parameter recovery error for the input-related LDS parameters as well as incorporating model-comparison metrics for the model as a whole. To our knowledge, neither of these analyses has been conducted on LDS models with non-Gaussian emissions. Lastly, we demonstrate the benefits of this method when applied to the behavior of mice performing a perceptual decision-making task.

## 2 Background and related work

### 2.1 Applications of state-space models in neuroscience

LDS and related state-space models have a long history in neuroscience. Bernoulli- and Poisson-LDS models have been particularly popular for inferring latent processes underlying neural spike trains (Gao et al., 2015; Nonnenmacher et al., 2017; Zoltowski et al., 2020; Valente et al., 2022). LDS models are also useful for linking neural dynamics to animal behavior (Linderman et al., 2019), while discrete state-space models like hidden Markov models (HMMs) have become increasingly common tools for characterizing both neural (Escola et al., 2011) and behavioral data (Wiltschko et al., 2015; Calhoun et al., 2019; Wiltschko et al., 2020). More recently, there has been growing interest in using state-space models to describe behavior: (1) using continuous rather than discrete variables (Johnson et al., 2016; Costacurta et al., 2022); (2) specifically in two-alternative forced-choice (2AFC) decision-making contexts (Roy et al., 2021); and (3) while understanding the effects of inputs such as sensory stimuli (Calhoun et al., 2019; Bolkan et al., 2022; Ashwood et al., 2022). Bernoulli-LDS models lie at the intersection of all three objectives, and thus their utility (along with methods that make their inference more efficient) will likely be central to neuroscience research in the future.

### 2.2 Extensions to LDS models

In addition to standard approaches, there has also been substantial work extending LDS models to more flexible dynamical systems in order to better capture important features in neuroscience data. For example, Gaussian-process factor analysis (GPFA) was developed to provide insight into neural signals by extracting smooth, low-dimensional trajectories from recorded activity (Yu et al., 2009). Building on GPFA, Orthogonal stochastic linear mixing models (OSLMMs) relax the assumption that correlations across neurons are time-

invariant and provide innovations in the regression framework that makes inference more tractable for large datasets (Meng & Bouchard, 2022). Other approaches employ LDS models as constituents of multi-component frameworks in order to capture more complex temporal activity patterns in neural data. One class of examples includes latent factor analysis via dynamical systems (LFADS) and its extensions, which use recurrent neural networks (RNNs) to recover neural population dynamics (Pandarinath et al., 2018; Prince et al., 2021; Zhu et al., 2022). Another approach employs variants of switching linear dynamical systems (sLDS), which learn flexible, non-linear models of neural and behavioral data by treating activity as sequences of repeated dynamical modes (Linderman et al., 2019; Nassar et al., 2019; Zoltowski et al., 2020; Mudrik et al., 2022; Wang et al., 2022; Weinreb et al., 2023). These advances make clear that LDS models serve a valuable and fundamental purpose in neuroscience and will continue to act as an essential building block for future work.

## 2.3 Inference methods

Fitting an LDS (as well as any of the extensions mentioned above) requires inferring the parameters governing the latent dynamics and emissions of the system. The most common approaches to inference are the Expectation Maximization (EM) algorithm (Baum et al., 1970; Dempster et al., 1977; Shumway & Stoffer, 1982) and Markov Chain Monte Carlo (MCMC) methods (Metropolis et al., 1953; Hastings, 1970; Geman & Geman, 1984; Gelfand & Smith, 1990). MCMC is a simulation method that produces samples that are approximately distributed according to the posterior; these samples are then used to evaluate integrals once convergence is reached. One of the benefits of MCMC is that it tends to work even for complicated distributions. However, it is often difficult to assess accuracy and evaluate convergence. MCMC is also slow and tends to require a large number of samples. EM, on the other hand, is an iterative optimization technique that returns a local maximum of the likelihood. EM is advantageous for a number of reasons, including that the "M-step" equations (in which parameter values are updated) often exist in closed form, the value of the likelihood is guaranteed to increase after each iteration of the algorithm, and in many cases EM requires fewer samples than MCMC. Several flexible variants of EM also exist. For example, when the conditional expectation of the log-likelihood is intractable, it is possible instead to use numerical approaches or to compute an analytical approximation using a technique such as Laplace's method (Steele, 1996) or variational inference (Blei et al., 2017). In this case, optimization occurs over an "Evidence Lower Bound" (ELBO) of the likelihood of the observed data. Despite these benefits, one major drawback of EM is that it is only guaranteed to find a local maximum. Thus, it often takes multiple initializations (or else a smart choice of initial parameters) to effectively find the global maximum. For this reason, methods such as bestLDS that identify good initializations for EM stand to greatly improve its computational efficiency. This is in contrast to MCMC, wherein it is typical to discard the first several samples (due to being poor representations of the posterior distribution) and therefore the initialization scheme is less important.

## 2.4 Initialization approaches for EM

Given the impact that the choice of initialization has on EM performance, it's no surprise that there is an extensive body of work on "smart" EM initialization methods for a variety of state-space models, including both LDS models (Buesing et al., 2012; Hazan et al., 2017; 2018) and HMMs (Siddiqi et al., 2010; Hsu et al., 2012; Anandkumar et al., 2014; Liu & Lemeire, 2017; Mattila et al., 2017). Often, these techniques rely on a combination of moment-based estimators (Martens, 2010; Buesing et al., 2012; Hsu et al., 2012; Anandkumar et al., 2014; Mattila et al., 2017) and subspace identification methods (Ho & Kalman, 1966; van Overschee & de Moor, 1994; Viberg, 1995; van Overschee & De Moor, 1996; Andersson & Rydén, 2009; Qin, 2006). Despite this work, a significant gap has persisted in that these methods (1) often don't account for input-driven data and (2) do not work for binomial distributions. BestLDS addresses both of these shortcomings. Although there have been several developments for augmented approaches to EM in logistic models using Polya-Gamma latent variables (Polson et al., 2013; Schein et al., 2016), these methods address other inference challenges in binomial-distributed data rather than improvements in the initialization scheme. However, it will be an interesting case for future study whether bestLDS can be combined with these methods to further improve inference for logistic models.

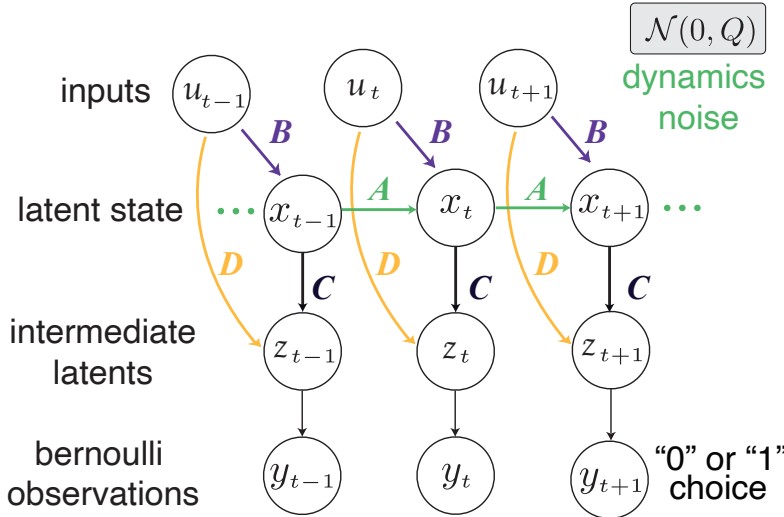

Figure 1: **Generative model schematic.**

## 3 Theory

### 3.1 Model description

Let $y_t$ denote the $q$-dimensional observation at time $t$. Concretely, we assume that $y_t$ is emitted as Bernoulli noise from a latent dynamical system with $p$-dimensional linear-Gaussian dynamics, potentially driven at each time-step by an $m$-dimensional input $u_t$. That is:

$$
\begin{aligned}
x_0 &\sim \mathcal{N}(\mu_0, Q_0) \\
x_t \mid x_{t-1} &\sim \mathcal{N}(Ax_{t-1} + Bu_t, Q) \\
z_t &= Cx_t + Du_t \\
y_t \mid z_t &\sim \text{Bernoulli}(f(z_t))
\end{aligned}
\tag{1}
$$

where $\mu_0, Q_0$ parameterize the initial distribution for the latent variable, $A$ and $B$ refer to the autoregressive and input-driven components of the mean of the latent dynamics, and $Q$ is the covariance of the latent Gaussian noise. The quantity $z_t$ is a latent convenience variable describing the state of the LDS in the $q$-dimensional output space as a linear transformation of the latent state by the loadings matrix $C$ and the input-output interactions matrix $D$. Finally, the outputs are sampled from a Bernoulli distribution with mean $f(z_t)$ for an arbitrary function $f$ (generally taken to be either the logistic or probit function). (Figure 1 shows a model schematic.)

### 3.2 Subspace identification

In general, subspace identification algorithms operate by first extracting an estimate of the latent state and then using linear regression to identify the system parameters. Concretely, given the latent state sequence $(x_1, ..., x_N)$ and the inputs $(u_1, ..., u_N)$ one may identify $A$ and $B$ by least-squares regression, using $(x_1, ..., x_{N-1})$ and $(u_1, ..., u_{N-1})$ to predict $(x_2, ..., x_N)$. Furthermore, given $(x_1, ..., x_N), (u_1, ..., u_N)$, and the emissions $(z_1, ..., z_N)$, one can again use least-squares to compute $C$ and $D$.

In order to achieve this, subspace identification algorithms typically operate on quantities called block-Hankel matrices, consisting of time-lagged subsequences of input or ouput data. For example, in the linear-Gaussian

case, the input and output[1] block-Hankel matrices are defined as:

$$Z_{0|2k-1} \equiv \begin{pmatrix} z_0 & z_1 & \dots & z_{N'-1} \\ z_1 & z_2 & \dots & z_{N'} \\ & & \vdots & \\ z_{2k-1} & z_{2k} & \dots & z_{2k+N'-2} \end{pmatrix} \equiv \begin{pmatrix} Z_{0|k-1} \\ Z_{k|2k-1} \end{pmatrix} \equiv \begin{pmatrix} Z_p \\ Z_f \end{pmatrix}$$

$$U_{0|2k-1} \equiv \begin{pmatrix} u_0 & u_1 & \dots & u_{N'-1} \\ u_1 & u_2 & \dots & u_{N'} \\ & & \vdots & \\ u_{2k-1} & u_{2k} & \dots & u_{2k+N'-2} \end{pmatrix} \equiv \begin{pmatrix} U_{0|k-1} \\ U_{k|2k-1} \end{pmatrix} \equiv \begin{pmatrix} U_p \\ U_f \end{pmatrix}$$

Note that each $z$ and $u$ above is typically a (column) vector, so these are *block* matrices. The top half of $Z_{0|2k-1}$ and $U_{0|2k-1}$ (i.e., containing rows 0 through $k-1$) represent the "past" relative to the $k$th row; for simplicity we denote these $Z_p$ and $U_p$, respectively. The bottom half of each matrix, by contrast, represents the "future", which we denote by $Z_f$ and $U_f$, respectively. Note that if $N$ is the total number of data points, $N' = N - 2k + 2$ is the number of columns of the matrix. Finally, the time lag parameter $k$ may be called the Hankel size and is defined by the user with only the requirement that $k \geq p$ (Katayama, 2005). An in-depth description of this is beyond the scope of this article, but it can be shown that only the top $p$ singular values of the Hankel matrix are greater than zero. Correspondingly, a typical heuristic is to build the Hankel matrix with large initial $k$ and then look at its singular value spectrum to pick a smaller $k$ more reflective of the underlying dynamics.

In order to see how the block-Hankel matrix relates to the system parameters, we repeatedly expand the linear-Gaussian components of Eq. 1 to yield:

$$Z_p = \Gamma_k \begin{pmatrix} x_0 & \dots & x_{N'-1} \end{pmatrix} + \Psi_k U_p$$

$$\Gamma_k = \begin{pmatrix} C \\ CA \\ CA^2 \\ \vdots \\ CA^{k-1} \end{pmatrix} \qquad \Psi_k = \begin{pmatrix} D & 0 & \dots & 0 \\ CB & D & \dots & 0 \\ \vdots & & & \vdots \\ CA^{k-2}B & \dots & CB & D \end{pmatrix}$$

where $\Gamma_k$ and $\Psi_k$ are referred to as the extended observability and controllability matrices, respectively. An analogous relationship holds for $Z_f$ and $U_f$.

Different subspace identification algorithms are characterized by their approach to decomposing the Hankel matrix in order to extract the system parameters. In our case, we will use the N4SID algorithm (van Overschee & de Moor, 1994). Briefly, this involves computing the RQ decomposition of the data matrix:

$$\begin{pmatrix} U_p \\ U_f \\ Z_p \\ Z_f \end{pmatrix} = RQ^T \tag{2}$$

It can be shown that processing various sub-blocks of the matrix $R$ permits extraction of the latent state sequence or the observability/controllability matrices, either of which can then be used to recover each of the system parameters (Ho & Kalman, 1966; van Overschee & de Moor, 1994; van Overschee & De Moor, 1996; Qin, 2006).

---

[1] Here we overload $z$ from Eq. 1 to refer to the output, as $z$ can be seen as the emission of a linear-Gaussian LDS.

### 3.3 Moment conversion

In the Bernoulli case we do not have access to $\{z_t\}$, the "intermediate emissions" that arise from the linearly transformed latents and therefore cannot directly construct $Z_p$ or $Z_f$ as described above. However, we *are* able to infer their moments from the moments of $Y_p$ and $Y_f$ (defined analogously), which has been shown in the general case to be a sufficient proxy for use in SSID methods (Buesing et al., 2012).

In particular, let $z_p$ denote $(z_0, ..., z_{k-1})^T$ and $z_f$ denote $(z_k, ..., z_{2k-1})^T$, and define $u_p, u_f$ analogously. Then the covariance of these concatenated matrix-valued random variables has a form that allows us to recover $R$ in Eq. 2 via Cholesky decomposition:

$$\Sigma = \mathrm{cov}\left[\begin{pmatrix} u_p \\ u_f \\ z_p \\ z_f \end{pmatrix}\right] = RR^T. \tag{3}$$

Here, we are exploiting the fact that while the observations are not Gaussian, the intermediate emissions $z$ and their time-lagged representations $z_p$ and $z_f$ are Gaussian. This requires us to place an assumption of normality on the inputs, but we will show empirically that parameter recovery is good even when this assumption is violated (see Section 4.2). For now, we begin by describing how to convert the moments of $y$ in order to infer $\Sigma$.

The core goal of the moment conversion process is to obtain the mean and covariance of $(z_p, z_f)$ from the mean and covariance of $(y_p, y_f)$. In particular, since $z_p, z_f, u_p,$ and $u_f$ are all jointly normal, it is our goal to find their joint mean and covariance given the moments of $y_p, y_f$. For convenience, we overload the notation $u = \begin{pmatrix} u_p \\ u_f \end{pmatrix}$, $z = \begin{pmatrix} z_p \\ z_f \end{pmatrix}$, and $y = \begin{pmatrix} y_p \\ y_f \end{pmatrix}$. Then the quantities we are interested in are:

$$\mu = \begin{pmatrix} \mu^u \\ \mu^z \end{pmatrix} \qquad \Sigma = \begin{pmatrix} \Sigma^{uu} & \Sigma^{uz} \\ \Sigma^{zu} & \Sigma^{zz} \end{pmatrix} \tag{4}$$

We begin by relating the moments of $y$ to the mean $\mu^z$ and the covariance $\Sigma^{zz}$ of $z$. The required integrals in the logistic-Bernoulli case are analytically intractable, therefore we instead consider the probit-Bernoulli case. In this setting we have the implication $y_i = 1 \iff z_i \geq 0$, and can therefore conclude:

$$\mathbb{E}[y_i] = P(z_i \geq 0) = 1 - \Phi(0|\mu_i^z, \Sigma_{ii}^{zz})$$
$$\mathbb{E}[y_i y_j] = P(z_i \geq 0 \wedge z_j \geq 0) = 1 - \Phi_2(0|\mu_i^z, \mu_j^z, \Sigma_{ii}^{zz}, \Sigma_{jj}^{zz}, \Sigma_{ij}^{zz}) \tag{5}$$

where $\bullet_i$ refers to the $i$th entry of $\bullet$, and $\Phi$ and $\Phi_2$ denote the univariate and bivariate cumulative normal distributions, respectively. Unfortunately, since the diagonal second moments are degenerate (that is, $\mathbb{E}[y_i y_i] = \mathbb{E}[y_i]$), this constitutes an underdetermined system of equations characterizing $\mu^z$ and $\Sigma^{zz}$. To resolve this identifiability issue, we set the diagonal covariances $\Sigma_{ii}^{zz} = 1$ without loss of generality and then solve for $\mu_i^z$ and $\Sigma_{ij}^{zz}$ using a numerical root-finder.

In regimes with extremely high-autocorrelation, it is possible that the system in Eq. 5 will not yield a solution. In such cases, it is possible to convert these constraints into a minimization problem and search for the covariance that most closely matches the target. In practice, however, we expect that such regimes (where the vast majority of the data is comprised of either only 0s or only 1s) are rare in real-world applications and did not encounter the need to do so in our analyses.

Next, we turn our attention to the inputs. In our case, $\mu^u$ and $\Sigma^{uu}$ are immediately available as the empirical mean and covariance of the inputs. As such, it only remains to compute the cross-covariance $\Sigma^{uz}$. To that

end, we evaluate the cross-moment[2]:

$$
\begin{aligned}
\mathbb{E}[y_i u_j] &= \int_{-\infty}^{\infty} \int_0^{\infty} u p(u,z) \, dz \, du \\
&= \int_0^{\infty} \left( \int_{\infty}^{\infty} u p(u|z) \, du \right) p(z) \, dz \\
&= \int_0^{\infty} \left( \mu^u + \Sigma^{uz} {\Sigma^{zz}}^{-1} (z - \mu^z) \right) p(z) \, dz \\
&= \left( \mu^u - \Sigma^{uz} {\Sigma^{zz}}^{-1} \mu^z \right) \left( 1 - \Phi(0|\mu^z, \Sigma^{zz}) \right) + \Sigma^{uz} {\Sigma^{zz}}^{-1} \int_0^{\infty} z p(z) \, dz
\end{aligned}
\tag{6}
$$

The integral in the second term is the mean of a truncated normal distribution and can be rendered as (Korotkov & Korotkov, 2020):

$$
\int_0^{\infty} z p(z) \, dz = \frac{\sqrt{\pi} \beta \left( \mathrm{erf}(\beta) - 1 \right) + \exp(-\beta^2)}{2\alpha^2 \sqrt{2\pi \Sigma_z}}
$$

with $\alpha = 0.5 {\Sigma^{zz}}^{-1}$ and $\beta = -0.5 \mu_z {\Sigma^{zz}}^{-1}$. From this point, $\Sigma$ is converted to $R$ as described in Eq. 3 and given as an input to N4SID.

To summarize, the steps of bestLDS are:

1. Compute the moments of $y$

2. Convert those moments to moments of $z$ as defined in Equation 4

    (a) Compute $\mu$ and $\Sigma^{zz}$ by solving the system in Equation 5
    (b) Use $\mu$ and $\Sigma^{zz}$ to compute $\Sigma^{uz}$ by solving the system in Equation 6

3. Cholesky decompose $\Sigma = RR^T$

4. Take $R$ as the input to the standard N4SID algorithm to recover the system matrices

## 4 Results

### 4.1 bestLDS infers correct parameters on simulated datasets

We begin by illustrating the effectiveness of the moment conversion for a very high-dimensional system (observation dimension $q = 30$, latent dimension $p = 15$, Hankel parameter $k = 20$, and input dimension $m = 3$). We sampled from a probit-Bernoulli LDS model and display the sample covariance of the observations, $\mathrm{cov}[y_t, y_{t+1}]$, the transformed covariance of the latents obtained by moment conversion, $\mathrm{cov}[z_t, z_{t+1}]$, as well as the true latent covariance matrix (Figure 2a). Observe that the output covariance does not match the latent covariance, while the converted moments do.

Next, we demonstrate the relative insensitivity of our algorithm's performance to the Hankel size $k$ by examining the singular value spectrum of the Hankel matrix, which one can use to determine the proper choice of the latent dimension $p$ for independent analysis or prior to using a fitting method like EM (see section 4.3). For simulated data of $N = 256,000$ in which $p = 5$, we repeatedly fit bestLDS while varying the Hankel size ($k = [3, 5, 7, 10]$). The results show that the general structure of of the singular value spectrum is independent of $k$ (Figure 2b). Indeed for all four simulations only the first five singular values are non-zero, consistent with the true structure of the data.

To illustrate the effectiveness of bestLDS, we next show that the estimator returns consistent, accurate estimates of the generative parameters on two simulated datasets, one low-dimensional and the other high-dimensional. In both datasets, ground-truth data is drawn by simulation from a probit-Bernoulli LDS with

---

[2]For readability, we omit the subscripts $i$ and $j$ to the right of the equals sign, but each $z$ is really $z_i$ and each $u$ is really $u_j$

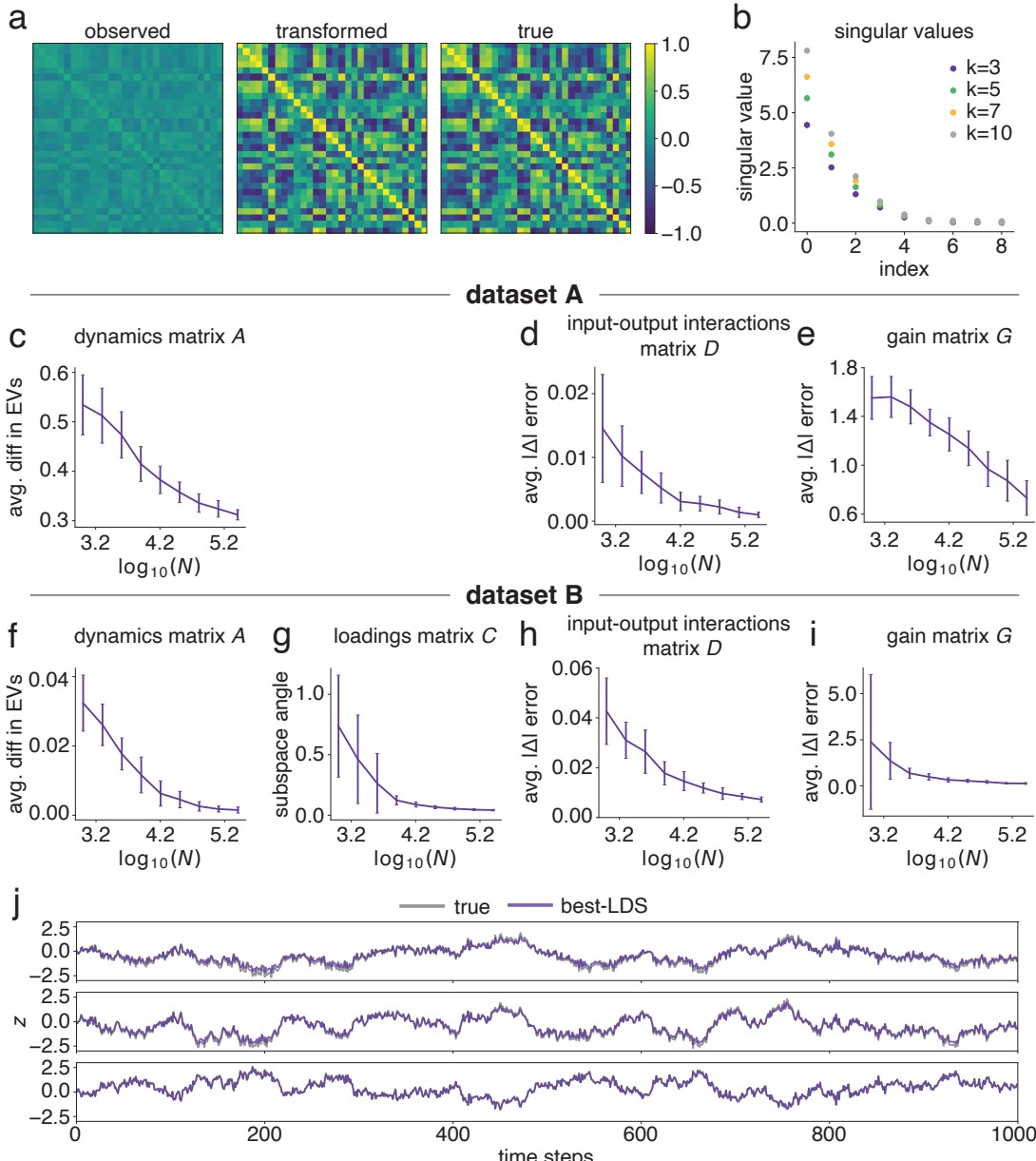

Figure 2: **Spectral estimator recovers simulation parameters in a variety of settings.** **(a)** Covariance matrix $\mathrm{Cov}[y_{t+1}, y_t]$ showing the sample covariance of the binary observations $y$ (left), the covariance of $z$ obtained by converting the moments of $y$ (middle), and the ground truth covariance of $z$ (right) for a simulated dataset ($q = 30$, $p = 15$, $k = 20$, $m = 3$). Note that the transformed covariance closely matches the ground truth. **(b)** The singular value spectrum of the Hankel matrix computed at various settings of the Hankel size $k$. For all settings, only the top $p = 5$ singular values are greater than zero. **(c-e)** Recovery error for bestLDS inferred parameters vs. training data size for a low-dimensional dataset with $k = 3$. For (c) we use the average absolute difference in the eigenvalues as the recovery error, and for (d) and (e) we use the average absolute elementwise difference. **(f-i)** Recovery error for bestLDS inferred parameters vs. training data size for a high-dimensional dataset with $k = 10$. For (g) the recovery error is the angle of the subspaces spanned by $C$ and $\hat{C}$. **(j)** Latent state trajectories simulated from true and inferred parameters for one of the simulations in dataset B. Both datasets exhibited strong temporal correlations and small interaction terms between the latents/outputs and the inputs. See Appendix for more details.

Table 1: **Comparison of the mean elementwise error in the gain matrix for inferred and true parameters across three different datasets.** Data was simulated using a trial structure, with five total trials. Inference was performed on training sets composed of four trials aggregated together, and reported error values are the means across those training sets. ±values indicate the standard error of the mean. The number of data points listed in the table corresponds to the full size of the dataset (all five trials). See Appendix for details on simulation parameters.

| Model Comparisons | | | | | | |
|---|---|---|---|---|---|---|
| **Method** | **Dataset A*** | | **Dataset B** | | **Dataset C** | |
| | **50k** | **256k** | **50k** | **256k** | **50k** | **256k** |
| **bestLDS** | 3.79 ±1.25 | 3.71 ±1.25 | 0.30 ±0.03 | 0.19 ±0.01 | 0.20 ±0.01 | 0.17 ±0.01 |
| **pLDS** | N/A | N/A | 2.37 ±0.90 | 2.30 ±0.28 | 1.17 ±0.11 | 0.63 ±0.03 |
| **gaussLDS** | 4.65 ±1.25 | 3.90 ±1.16 | 2.45 ±0.02 | 2.25 ±0.01 | 3.02 ±0.25 | 2.24 ±0.12 |

$\mu_0 = 0$ and $Q_0$ taken to be the stationary marginal covariance of the latents. $A, B$, and $D$ are sampled independently from a standard normal distribution and then have their eigenvalues thresholded (spectral radius $< 1$ is needed for $A$ to ensure stability). We take $C$ to be a random orthonormal matrix such that, given $A$, the stationary marginal covariance of $z$ has unit diagonal (results of simulations where we do not make this assumption on the marginal covariance are available in Supplementary Figure A.1).

Note that LDS models are not uniquely identifiable and SSID algorithms in general recover parameters only up to a similarity transformation. Concretely, let $x' = Tx$ for $T$ an invertible matrix. Then we may recover $A' = T^{-1}AT, B' = T^{-1}B$ and $C' = CT^{-1}$. $D$ is recovered canonically and thus we can look at the mean absolute elementwise error to establish the accuracy of our estimator for this matrix. For the other parameters, however, we must establish alternative error metrics. For $A$, we measure the error of its estimate $\hat{A}$ by examining the average absolute difference in their eigenvalues. For $C$, we look at the angle between the subspaces spanned by the true $C$ and the estimate $\hat{C}$. For $B$, there is not a simple error metric. However, we can examine the gain matrix $G = C(I - A)^{-1}B + D$, which is preserved canonically under the similarity transformation, taking the mean absolute difference across the inferred and true gain matrices as an overall indicator of accuracy in the recovery process. For both datasets, we computed these error metrics and took their average across 30 simulations for $N \in [1000, 256000]$. The time taken to run bestLDS on these datasets is available in Supplementary Figure A.2.

The accuracy of the system matrices recovered via bestLDS are displayed in Figure 2c-i. In the low-dimensional dataset (dataset A), the errors for each relevant error metric decrease with increasing data size (note subspace angle is not possible to compute for $C$ when $q < p$). While the errors for $A$ and $G$ are non-zero even at $N = 256000$, systems in which $q < p$ (i.e. high-dimensional latent structure governs low-dimensional outputs), are a particular challenge for LDS models. Therefore it is notable that we nonetheless see decreasing errors—and for certain metrics quite small errors—in this regime. As a contrasting example, we simulated a high-dimensional dataset with higher-dimensional latent dynamics (dataset B). Our estimator performs extremely well in this regime. For the system matrices $A$, $C$, and $D$ as well as the gain matrix $G$ (Figure 2f-i), the error metrics decay to zero, indicating that our estimator is consistent. Furthermore, the errors are quite low across the whole range of dataset-sizes taken—for example, the mean error in $A$ at $N = 1000$ is less than 0.05. For completeness, we compared the accuracy of our estimator against running N4SID directly on $z$ (i.e., pretending we have access to the Gaussian subset of the LDS and therefore forgoing the need to perform moment conversion) and found that bestLDS did not perform significantly worse on most variables (Supplementary Figure A.3). Thus our estimator is quite effective, even in a sparse-data regime. To further validate our parameter recovery for dataset B, we simulated noiselessly using both the true and inferred parameters and found that the latent state traces overlapped almost exactly (Figure 2g).

Lastly, we performed model comparisons on three datasets of different sizes to illustrate the accuracy of our recovery process in different settings (Table 1). Dataset B has the same simulation parameters as in Figure 2. Dataset C has lower autocorrelations, larger instantaneous correlations, and similarly small input-latent/output interactions. Dataset A* is the average over five datasets with the same generative parameters as dataset A; we did this to make our model comparison results more robust given the peculiarities of the $q < p$ regime. Focusing on the gain matrix, as it is uniquely identifiable and incorporates the major

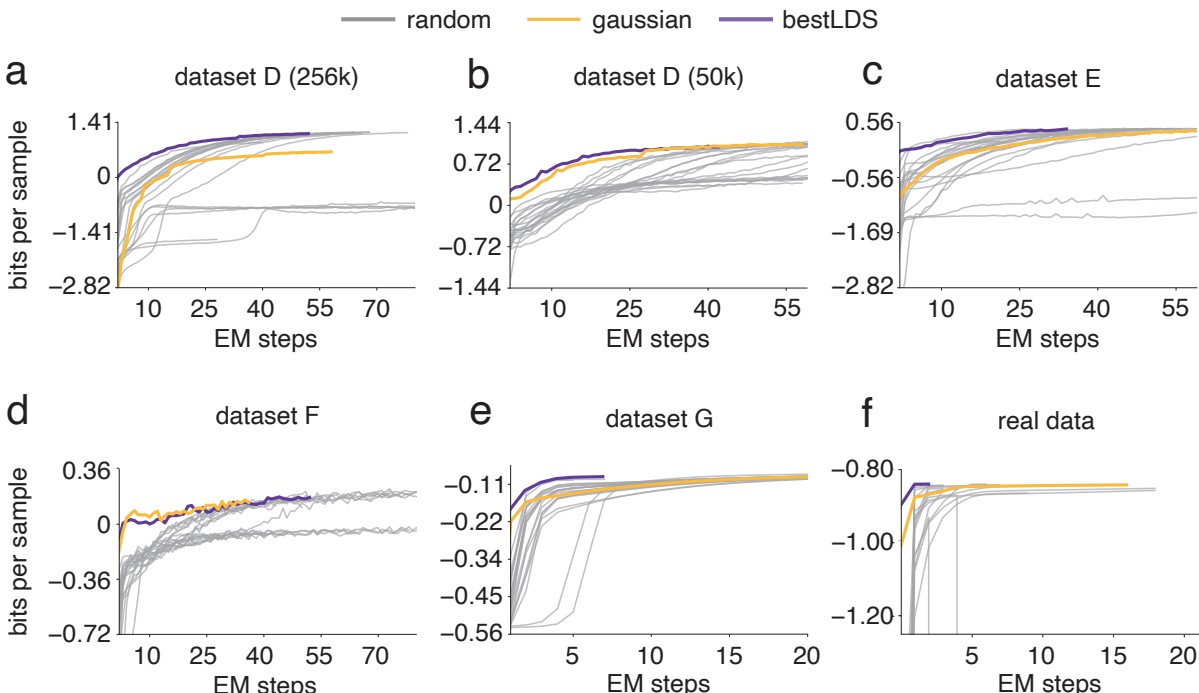

Figure 3: **bestLDS is an effective initialization for EM.** For panels a-d, the ELBO (converted to bits/sample) is plotted on each step of EM with random initializations, a bestLDS initialization, and a Gaussian initialization. Panels e-f are the same except that EM returns the log-evidence at each step. **(a)** Dataset D is characterized by slow rotational dynamics and high-magnitude loadings matrix. It exhibits long stretches of 0s and 1s. Inputs were drawn from a zero-mean normal distribution with covariance $\Sigma_u = 10^{-4}I$. **(b)** Same as (a) but with $N = 50,000$. **(c)** Dataset E is characterized by extremely fast rotational dynamics. It exhibits switching behavior, alternating emissions of 0s and 1s. Inputs were drawn from a zero-mean normal distribution with covariance $\Sigma_u = 0.1I$. **(d)** Dataset F is similar to dataset B but has inputs sampled from a Student-t distribution with three degrees of freedom. **(e)** Dataset G has one-dimensional observations ($q = 1$) with slow rotational dynamics. See Appendix for more details on all simulation parameters. **(f)** Data taken from mice performing a two-alternative forced choice task in virtual reality. See Section 4.3 for details.

parameters of the system, we computed the mean absolute elementwise error between the true generative parameters and three sets of inferred parameters: the bestLDS inferred parameters, the inferred parameters using the Poisson LDS (pLDS) estimator as detailed in (Buesing et al., 2012), and the inferred parameters using a linear-Gaussian (gaussLDS) estimator[3]. In the cases we have examined, bestLDS achieves the smallest gain error amongst the benchmarks tested, and in many instances those errors are extremely low. This indicates that there are conditions in which the bestLDS parameters can suffice on their own in capturing the characteristics of the system even without subsequent optimization. This also demonstrates the need for estimators that account for the specific distribution of the underlying data, as it is not sufficient to simply substitute a different estimator and achieve accurate results.

## 4.2 bestLDS provides good initializations for EM

In regimes where additional accuracy is desired, bestLDS may serve as a smart initialization method for EM. Indeed, estimates returned by spectral estimators have previously been shown to serve as good EM initialization points (Martens, 2010; Buesing et al., 2012). This is particularly important given that EM fitting procedures typically require the user to initialize randomly (or at least, without complete knowledge of what a good initialization would be) and therefore a single fit is rarely sufficient to determine that EM has

---

[3]computed by treating the binary data as real-valued and simply fitting a Gaussian LDS to the data using N4SID (i.e., skipping the moment conversion step)

Table 2: **Time to convergence for Laplace-EM for different initialization methods.** For random initializations ($N = 20$), both total and mean (total / mean) times are reported. For bestLDS initializations, we report both the time to convergence and the time to acquire the initializations using the bestLDS estimator (estimator time + convergence time). All computations were performed on an internal CPU cluster.

| Convergence Times for Laplace-EM (min.) | | |
|---|---|---|
| **Initialization** | **Dataset D** (256k) | **Dataset D** (50k) | **Dataset E** (256k) |
| **random** | 902 / 45.10 | 232.40 / 11.62 | 571.75 / 28.09 |
| **Gaussian** | 38.67 | 11.60 | 39.67 |
| **bestLDS** | 0.15 + 34.67 | 0.09 + 8.40 | 0.14 + 19.83 |
| **Initialization** | **Dataset F** (100k) | **Dataset G** (256k) | **Real Data** (54883) |
| **random** | 514.33 / 25.72 | 886.67 / 44.33 | 25.11 / 1.67 |
| **Gaussian** | 12.67 | 49.40 | 5.61 |
| **bestLDS** | 0.10 + 19.67 | 0.05 + 8.87 | 0.02 + 0.33 |

found the global optimum. Instead, users often fit EM to their data many times and take the parameters from those that produced the highest log-likelihood or ELBO (Evidence Lower Bound). This can be extremely time-consuming, especially when datasets are large. Thus, a method that can generate initializations that (1) converge quickly and (2) consistently converge to the global optimum can substantially reduce compute-time.

Accordingly, we examined how useful bestLDS is in such a role. In this section we assess the performance of this estimator as an initialization for Laplace-EM[4], an EM method using the Laplace approximation to efficiently compute the log posterior (Smith & Brown, 2003; Kulkarni & Paninski, 2007; Paninski et al., 2010; Yuan & Niranjan, 2010). We then compare the results to random and linear-Gaussian[5] initializations when fit to three different simulated datasets. For each dataset and initialization method, we have plotted the ELBO as a function of the number of EM steps taken until the system reaches convergence[6]—this is a useful metric to examine since our primary interest is training time and not performance. For the sake of clarity, we will briefly describe each dataset considered for this purpose and have provided a full accounting of all the generative parameters in the Appendix.

First, we examined the case in which the dimensionality of the outputs was greater than that of the latents (i.e., $q > p$). In particular, we considered datasets exhibiting properties that one might expect from typical Bernoulli datasets (e.g., mouse decision-making behavior). For example, dataset D is characterized by long stretches of 0s or 1s, which we might expect in behavioral datasets that exhibit relatively high autocorrelation (e.g., habitual behaviors, perseveration, etc.); this was implemented by imposing slow rotational dynamics. Here, the bestLDS initialization substantially outperforms Gaussian and random initializations, eliciting higher ELBOs from the first step of EM and converging to the optimal solution in fewer total steps (Figure 3a), thus saving notable computation time (Table 2, top, first column). This result also holds for the same dataset fit to fewer data points (Figure 3b). In this case, while the Gaussian initialization also achieves the global optimum, it still takes more time to converge than the bestLDS initialization and in fact offers negligible time-savings over the average random initialization (Table 2, top, second column). Dataset E reflects a different extreme of Bernoulli-distributed behavior in that it produces observations that rapidly switch between 0s and 1s, which we implement by imposing fast rotational dynamics. We might expect this type of data from alternation tasks (i.e., the subject must make the opposite choice from the previous trial). The bestLDS initialization significantly aids performance on this dataset as well (Figure 3c) and reduces convergence time over Gaussian initializations by approximately 50% (Table 2, top, third column). Finally, dataset F is almost identical to dataset B (see Figure 2 and Table 1) except that inputs are drawn from a Student-t distribution with three degrees of freedom, thus testing the flexibility of our assumptions on the input sampling distribution. Here we find that both non-random initializations perform well, indicating the general utility of these spectral methods on input-output data even when the inputs are non-Gaussian (Figure 3d and Table 2, bottom, first column).

---

[4]Implemented using the Python ssm package, licensed under the MIT license

[5]using the same process as described in the model comparisons section

[6]for the $q \geq p$ regime, we say that EM has converged when the avg. $|\Delta|$ in the gain matrix on successive steps is within a tolerance ($tol = 0.15$); for $p < q$ we found that comparing the log-evidence on successive steps to be more reliable ($tol = 10$, in bits/sample).

Next, we turned our attention to the $q < p$ case, specifically focusing on when $q = 1$ due to its practical relevance for neuroscience decision-making experiments, in which subjects are more likely to undergo tasks one-by-one and elicit a single choice output at a time. Dataset G is a simulated dataset in this regime, characterized by slow rotational dynamics. As earlier, the bestLDS estimate attains a higher log-evidence than comparative methods from the first step of EM (Figure 3e) and converges much faster than the Gaussian or random initializations (roughly a factor of 5.5 faster; see Table 2, bottom, second column). Finally, we performed these comparisons on binary data from mice performing a decision-making task (described in further detail in the next section). Here, the bestLDS initialization converges in only three steps of EM (Figure 3f). All other initializations take significantly longer to converge (on the order of 17x slower for Gaussian LDS and 76x slower for the random initialization total time; see Table 2, bottom, third column) and also have initial log-evidence values that are noticeably lower than the bestLDS estimate.

### 4.3 Application to mouse decision-making data

To asses the performance of our estimator in real-world settings, we turned to a dataset from mice performing a sensory decision-making task[7] (Bolkan et al., 2022). In this experiment, mice ($n = 13$) navigated a T-shaped maze in virtual reality. Mice were trained to detect visual cues that appeared to each side of their field of view while running down the main stem of the maze. At the end of the main stem, they had to turn toward the side of the maze with the most cues in order to receive a water reward (Figure 4a). The mice repeated this task for many trials in a row, with an average daily session consisting of approximately 200 trials ($N = 54,883$ total trials). On a random subset of 15% of trials within each session, the animals' striatal dopamine D2 receptor medium spiny neurons (MSNs) were inactivated optogenetically using a 532nm laser. This transient inhibition provides a mechanism for understanding how the mice perform the task when activity in their striatum, a brain region long associated with decision-making and motor output (Cox & Witten, 2019; Tang et al., 2022), is suppressed. In addition to its broad association with decision-making, the striatum is the chief target of dopamine in the brain and has previously been demonstrated to encode reward prediction error signals (Schultz et al., 1997). D2 MSNs in particular comprise a cell-type in the striatum that has specifically been linked with suppression of movement and constitute the first locus in what is known as the "indirect pathway" of the striatum (see (Cox & Witten, 2019) for a review).

When modeling the data from this task, we take the following as inputs $u_t$ to bestLDS, given their high likelihood as relevant predictors in this task: 1) the difference between the number of right and left cues on each trial, 2) the animal's previous choice if rewarded (coded in the model as $\pm 1$ if the mouse made a correct right/left choice and 0 otherwise), and 3) the presence of inhibition (coded as $\pm 1$ if delivered to the right/left hemisphere of the brain and 0 otherwise). We also consider including $n$-th order previous choices as additional inputs (coded as $\pm 1$ for the right/left choice the animal made $n$ trials in the past, regardless of reward status). For our preliminary analysis, we vary the total number of previous choices incorporated in the model for $n \in \{0, 1, 2, 3, 4\}$, yielding $m \in \{3, 4, 5, 6, 7\}$.

To determine our choice of $p$, we first examine the singular value spectrum of the transformed data (Figure 4b). For this dataset, increasing $p$ yields only marginal decreases in the singular value after $p = 4$. We correspondingly take $p = 4$ as an upper-bound and conduct further analysis within the range $p \in \{2, 3, 4\}$, while also varying the number of inputs as previously described. In particular, we fit bestLDS to the data and looked at how well the inferred parameters predict subsequent choice (Figure 4c-d) for each combination of $p$ and $m$. For all settings of $m$, the inferred parameters with $p = 3$ outperform the other $p$ settings and predict choice well ($> 70\%$). Of these, the best-performing setting is with $m = 4$ (1st-order previous choice). Therefore, for all remaining analyses we fit bestLDS with $m = 4$ and $p = 3$.[8]

It's worth noting that this approach for specifying *a priori* the dimensionality of the system showcases an additional benefit of bestLDS. Without our method, the typical approach would be to fit many variants of the model using a standard inference procedure like EM, choosing the state number by some comparative method (e.g., cross-validation). However, this tends to be quite time-consuming and computationally costly. Here, the singular value spectrum combined with bestLDS fitting immediately gives an informative summary

---

[7]Data obtained with permission directly
[8]Note that because $q = 1$, this places us in the $q < p$ regime, which can sometimes suffer from performance issues despite performing well here; see Figure 2.

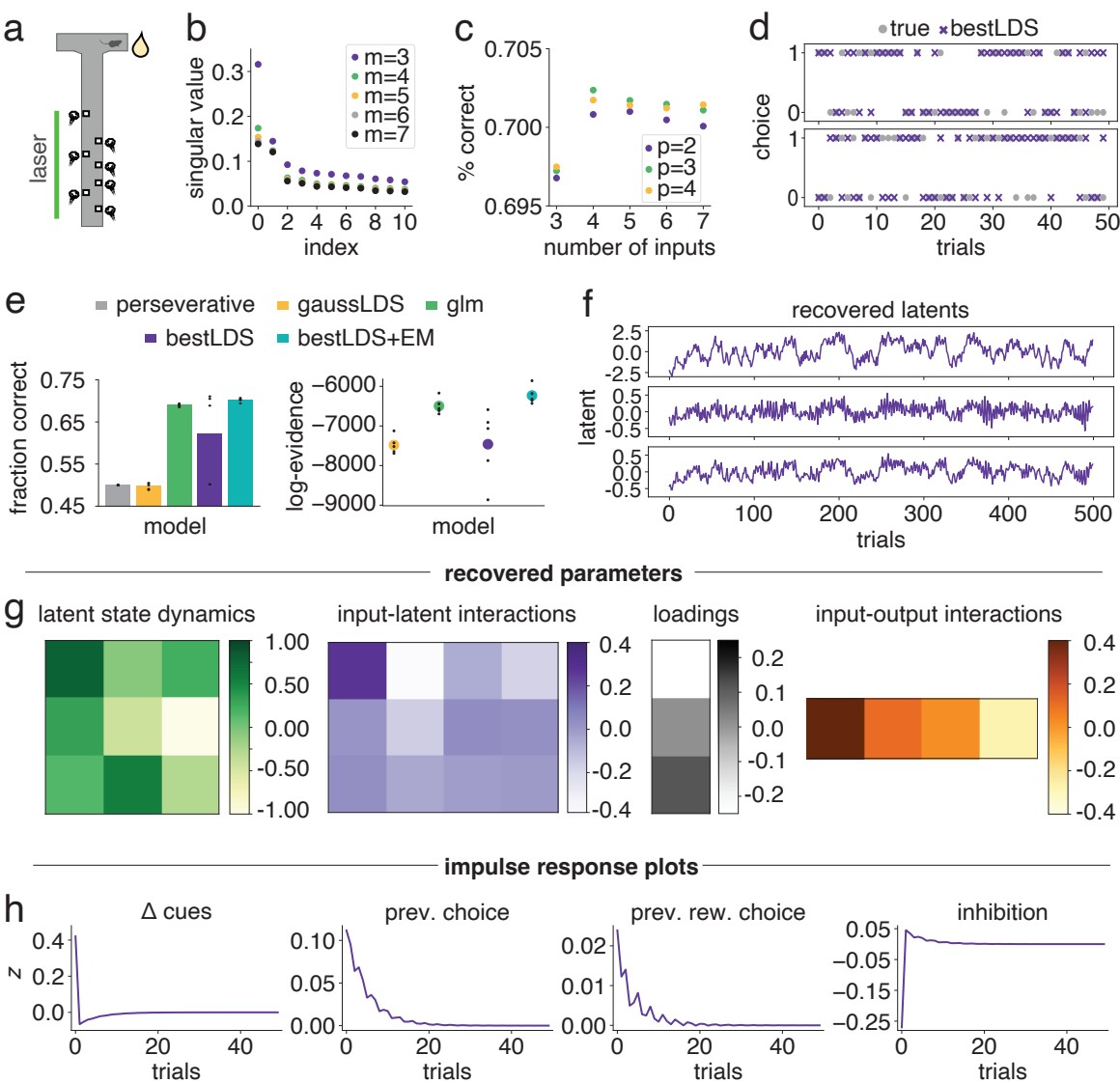

Figure 4: **Analysis of mouse binary decision-making data with bestLDS.** **(a)** Task schematic. **(b)** Singular value spectrum of the Hankel matrix with increasing number of previous choices used as inputs. **(c)** Prediction accuracy of bestLDS for various settings of $p$ and $m$. **(d)** Example stretches of predicted and true choices. **(e)** 5-fold cross-validated performance comparisons for a variety of methods on this dataset. Left: fraction of choices correctly predicted. Right: test set log-evidence. Black dots indicate each of the individual folds. **(f)** An example stretch of recovered latents. **(g)** The fitted parameters estimated by bestLDS + EM. **(h)** Impulse responses for the fitted system. In each subplot, data has been simulated noiselessly with a unit input in the indicated dimension.

of the parameters of the dataset, providing in mere minutes what would otherwise take many hours and hundreds of different EM initializations (that is, dozens of initializations per parameter setting and value). Our spectral estimator can even be applied for this purpose when the ultimate goal is to fit to a different state-space model, such as discrete-state HMMs.

To further assess bestLDS performance on the real data, we use five-fold cross-validation to compare its performance to three other models: a simple perseverative model, which predicts a high probability (70%) of making the same choice as on the previous trial; a Gaussian LDS[9]; and a Bernoulli generalized linear

---

[9]We convert the real-valued observations $z$ produced by the Gaussian LDS to binary observations by taking $y_t = 0$ if $z_t < \bar{z}$ and $y_t = 1$ if $z_t >= \bar{z}$

model (GLM; Figure 4e). Specifically, we operationalize "performance" in two ways: fraction of choices in the test set correctly predicted and the log-evidence of the data evaluated under the fitted parameters. Under both metrics, bestLDS significantly outperforms the perserverative and Gaussian LDS models. Furthermore, four out of five test sets achieve equal or greater performance than the GLM on the prediction accuracy metric, though this is not borne out in the log-evidences. This is already notable, given that bestLDS only returns an estimate of the system parameters for a multi-state LDS, whereas the GLM is a regression model that is capable of finding fairly accurate weights for a dataset of this size. However, when coupled with EM, which converges in just three steps and takes less than a minute to run, we find that the resultant inferred parameters perform better than all other methods regardless of metric.

Not only does bestLDS perform well against comparative models, but it also offers unique scientific insights into the data that don't require subsequently using the inferred parameters as EM initializations. For example, inspection of the recovered latents $x$ shows that one of the latent dimensions dominates the tendency of the overall response; the inferred traces on this latent dimension exhibit significantly higher magnitude than the other two (Figure 4f), and the corresponding weights in the dynamics matrix $A$ are also of high magnitude (Figure 4g, left column). As such, "activity" in this dimension can be described as strongly driving both the decisions of the mouse and also the latent state of the LDS underlying it. We can examine how this is affected by the inputs by examining input-latent interactions matrix $B$; here we see that the largest magnitude elements of the matrix are those mapping the $\Delta$ cues and previous choice inputs onto the high-magnitude dimension. Overall, this suggests that the first latent dimension is mostly driven by autocorrelation (since its self-weight in $A$ is high and the activity of the other dimensions is low), with strong influence due to those inputs, and activity in this dimension largely dominates the contribution from the latent state to the outputs.

Furthermore, examining the input-output interactions matrix reveals the general effect that each input has on choice; more cues on the right side of the maze, a previous rightward choice, and a previous rightward rewarded choice are all more likely to elicit a rightward choice on the current trial (since they have positive weights, and right choices are coded as 1), whereas right-hemisphere inhibition is more likely to elicit a leftward choice (since it has a negative weight). Note that the results obtained from using the bestLDS-inferred parameters as an initialization for EM produce the same qualitative interpretation of the system, indicating that for this dataset it is possible to accurately characterize the relationship between the inputs, latent states, and animals' choices solely from bestLDS without using an additional inference procedure (see Supplementary Figure A.4). It's also worth noting that it would not be straightforward to reach the same conclusions by assuming linear-Gaussian observations and using standard N4SID methods. In our investigations, such an approach elicits uninterpretable system parameters, including large eigenvalues in $A$ and pathological impulse-response traces that diverge.

A convenient way to analyze the inferred parameters is to examine their impulse responses—that is, simulating a noiseless Bernoulli LDS with the inferred parameters starting with a unit input in a single dimension at time step 0 (Figure 4h). The resultant responses can be seen as a general summary of how each input affects the output state of the system—and have been shown to be fully sufficient to characterize LTI systems. For example, a +1 input in the $\Delta$ cues dimension (more right cues than left) makes the system more likely to emit a right-turn, as expected given the task structure. The second trace shows that the mice perseverate (i.e., are, on average, more likely to repeat their previous action than to do the opposite one), as a previous right choice more likely causes a right turn on the current trial. The same effect holds when the previous choice was rewarded, but with smaller magnitude, suggesting that perseveration is not strongly contingent on reward. The negative response in the final trace indicates that inhibiting neural activity in one hemisphere encourages the mice to turn contralaterally (i.e., inhibition of the striatum on the right hemisphere will influence the mice to turn left). Also note the relatively long decay times for choice-related inputs, which speak to the perseverative aspect of the dynamics. These conclusions largely match previously observed patterns in this data using input-output HMMs, and the performance of bestLDS+EM in predicting choice in this dataset is similar to that found in the discrete-state case (Bolkan et al., 2022). These findings raise interesting questions about the relationship between discrete- and continuous-state models for decision-making behavior that would be a promising avenue for future study.

## 5 Discussion

We have presented a spectral estimator for driven Bernoulli latent linear dynamical systems. On a variety of simulated datasets, bestLDS returns consistent, high-quality estimates of the generative parameters efficiently on its own and also as an initializer for the expectation-maximization algorithm. Estimates are particularly robust in the large data, high-dimensional regime, which is an especially desirable use case of the model given the potential time-savings in fitting. One notable limitation of the method is that accuracy of the estimates may suffer in low-dimensional regimes where the number of observations is smaller than the number of latent dimensions, however we show in Section 4.3 that predictive performance is still strong and the returned estimates yield novel scientific insights, in addition to the significant time-savings over traditional inference methods (i.e. EM). These results can then inform subsequent analyses and model fitting procedures that require longer compute time (e.g., an assessment of the optimal number of latent dimensions and the importance of different inputs on the latent dynamics). Identifying the system matrices in the continuous state case may also provide relevant information for discrete state-space models. This would be another interesting area of future work, given the popularity of such models of behavior in neuroscience (Bolkan et al., 2022; Ashwood et al., 2022).

In the extremely-high dimensional regime, the primary limitation is the need to numerically infer the moments of $z$. Since the number of parameters increases as $O\big((kq)^2\big)$, time spent on moment conversion grows quickly with the dimensionality of the data. However, the compute time required for moment conversion is unlikely to negate the time-savings garnered from faster EM convergence.

In sum, bestLDS proves an efficient estimator for driven Bernoulli-LDS data. Parameter recovery even without EM is efficient and accurate, potentially averting the need to spend time running iterations of expensive search algorithms. Coupled with EM, the method greatly speeds convergence of the final estimate, even in cases where normality of the inputs is violated. Binary time-series data arise in many contexts, including reinforcement learning and decision-making, weather outcomes, finance, and neuroscience. We thus expect that bestLDS will be broadly useful to practitioners looking to quickly acquire a description of the latent dynamics underlying these systems.

### Data availability

The data that support the findings of this study are publicly available on figshare at https://figshare.com/articles/dataset/bestLDS_associated_data/23750670.

### Code availability

Code for general use applications of bestLDS analyses developed in this study, including all applications to simulated and real data presented in the manuscript, are available on GitHub at https://github.com/irisstone/bestLDS/.

### Acknowledgments

I.R.S. was supported by the National Institute of Mental Health of the N.I.H. under award number F31MH131304. I.M.P was supported by an N.I.H. BRAIN initiative grant (9R01DA056404-04). J.W.P. was supported by grants from the Simons Collaboration on the Global Brain (SCGB AWD543027), the N.I.H. BRAIN initiative (9R01DA056404-04), and a U19 N.I.H.-N.I.N.D.S. BRAIN Initiative Award (5U19NS104648).

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

# A    Appendix

## A.1    Simulated dataset parameters

Table A.1: **Generative parameters for simulated datasets.**

| Simulated dataset parameters | | | | | | | | |
|---|---|---|---|---|---|---|---|---|
| **Dataset** | **A** | **B** | **C** | **D** | **Q** | **q** | **p** | **m** |
| **A** | eigenvalues in [.9,.99] | orthonormal scaled by .1 | orthornormal scaled by .1 | orthonormal scaled by .1 | $.1I$ | 1 | 3 | 3 |
| **B** | eigenvalues in [.9,.99] | orthonormal scaled by .1 | orthornormal scaled by .1 | orthonormal scaled by .1 | $.1I$ | 10 | 5 | 3 |
| **C** | eigenvalues in [.5, .9] | orthonormal scaled by .1 | orthonormal scaled by 10 | orthonormal scaled by .1 | $.1I$ | 8 | 6 | 4 |
| **D** | rotation matrix with $\theta = \pi/48$ | orthonormal scaled by .1 | orthonormal scaled by 1e4 | orthonormal scaled by .1 | $.0001I$ | 5 | 2 | 3 |
| **E** | rotation matrix with $\theta = \pi/2$ | orthonormal scaled by .1 | orthonormal scaled by 1e4 | orthonormal scaled by .1 | $.1I$ | 5 | 2 | 3 |
| **F** | eigenvalues in [.9,.99] | orthonormal scaled by .1 | orthornormal scaled by .1 | orthonormal scaled by .1 | $.1I$ | 10 | 5 | 3 |
| **G** | rotation matrix with $\theta = \pi/400$ | scaled by .01 | scaled by .25 | scaled by .2 | $.001I$ | 1 | 2 | 3 |

## A.2    Parameter recovery with non-unitized diagonal covariance

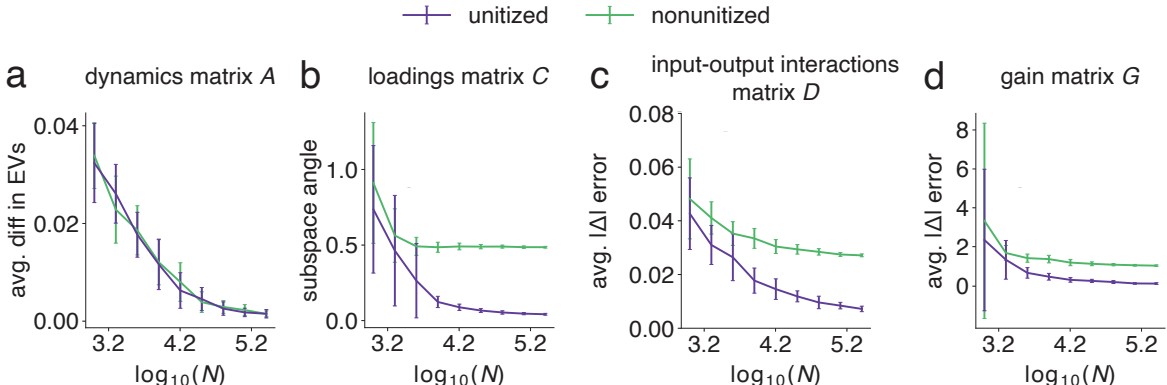

Figure A.1: **Stationary marginal covariance of $z$ having non-unit diagonal covariance affects inference of $C$ but mostly not other parameters. (a-d)** Average recovery error for bestLDS inferred parameters under unit-diagonal ("unitized") vs non-unit-diagonal ("nonunitized") regimes.

### A.3  Estimator run time as a function of dataset size

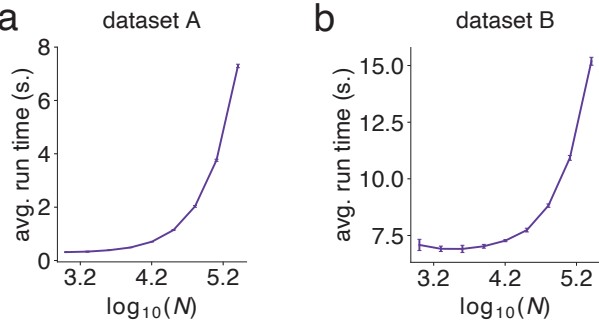

Figure A.2: **bestLDS run time as a function of dataset size.**

### A.4  Effect of moment conversion

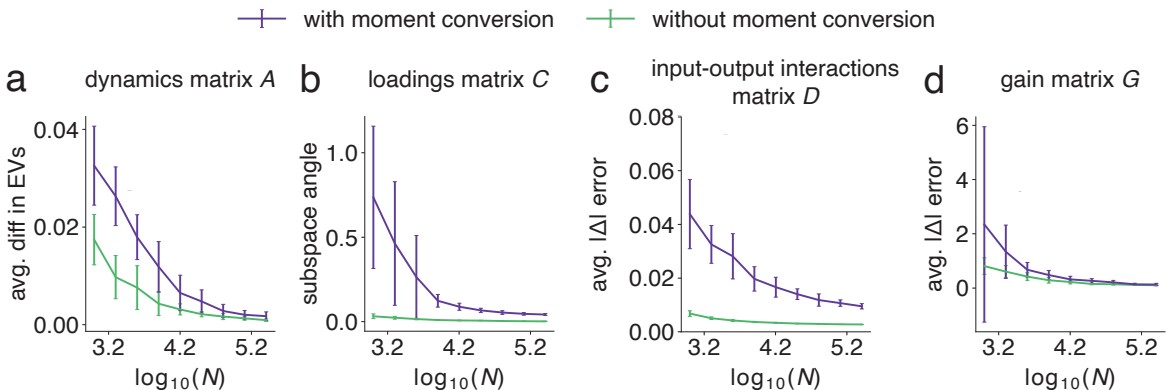

Figure A.3: **bestLDS underperforms compared to a model that has direct access to $z$. (a-d)** Average recovery error for bestLDS inferred parameters ("with moment conversion") vs. running N4SID directly on $z$ ("without moment conversion").

### A.5 Comparison of recovered system parameters for bestLDS and bestLDS+EM

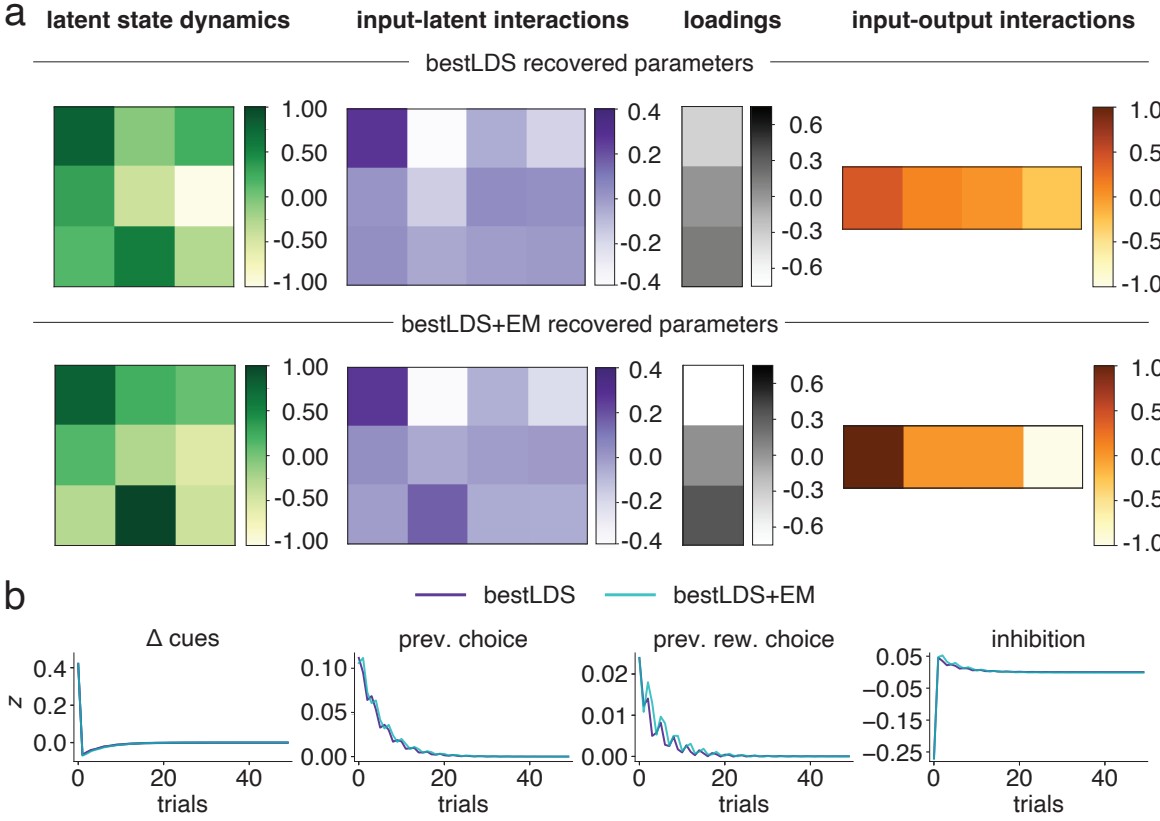

Figure A.4: **bestLDS recovered parameters closely match the inferred parameters using EM with bestLDS initializations.** **(a)** The recovered parameters using bestLDS (top) and bestLDS+EM (bottom). **(b)** Impulse responses for each of the fitted systems. In each subplot, data has been simulated noiselessly with a unit input in the indicated dimension.

