# OpenReview forum: "Spectral learning of Bernoulli linear dynamical systems models for decision-making"
_TMLR — Accepted by TMLR_

### Review · Reviewer_WsBP · 2023-05-11

**Summary Of Contributions:**

This paper proposes a spectral learning method for (possibly controlled) linear dynamical HMMs with Bernoulli observations, building on existing work for Gaussian/Poisson observations.The proposed method is fast, avoids hazards of local minima, and can be used to initialize EM algorithms for better solutions. The authors study their method on a number of synthetic tasks, and also apply it to analyze data from mice performing a decision-making task.



**Audience:**

Yes

**Broader Impact Concerns:**

I see no major broader impact concerns.

**Claims And Evidence:**

Yes

**Requested Changes:**

1) I think the paper would benefit with a clearer discussion of the technical novelty, in its current form it does not seem like a significant advancement over Buesing et al. The authors might also consider comparing with this approach.

2) I think the paper would benefit from a slightly clearer discussion of how the nonlinear systems of equations are solved. RIght now, it seems like this is a high-dimensional problem, but if I understand correctly, this is just a collection of uncoupled low-dimensional equations. It would also be nice if the paper also dwelt a little more on the situation when the equations don't have a solution, this is mentioned only very perfunctorily.

3) I would like to see some theoretical and empirical discussion of the cost of solving the nonlinear moment conditions.

4) I think "for decision-making" in the title is unnecessary and a bit confusing. The method also only applied for probit-transformations, which is not mentioned clearly upfront (in the abstract/introduction).


**Strengths And Weaknesses:**

Positives:
1) The problem is an important one in domains such as neuroscience, and the proposed solution seems useful and practical.
2) The paper is largely well written
3) The model shows good performance in the experiments, which are relatively thorough

Negatives:
1) As far as I can tell, this paper is a fairly straightforward extension of Buesing et al from the Poisson case to the Bernoulli case. The authors here use the same principle: the observations Y are a transformation of Gaussian intermediate outputs Z, and learning the parameters of the system involves inferring the mean-covariance structure of Z from Y. The main technical problem here is that for Bernoulli variables, since the mean equals the mean square, the system of equations is underdetermined. The authors solution is just to set the diagonal of the covariance of Z to be one and solve for the remaining covariance variables. I'm not convinced this is a significant methodological innovation. The paper also allows inputs u to the linear-dynamical system, but dealing with this is also a fairly immediate application of Buesing et al.

2) The proposed method only works for probit transformations, where the sysyem of nonlinear equations relating the moments of Y and Z can be written explicitly.

3) Evaluation metrics seem ad hoc and not well motivated. Why is recovery of A measured using eigenvalues, and C using the angle between the subspaces? I don't think the paper clearly enough discusses the effect on estimating C of fixing the diagonal of the covariance of z.

Questions:
1) Can we directly apply the estimator for the Poisson observations from Buesing et al, treating the Bernoulli outputs as outputs from a Poisson distribution with small rate parameter? Table 1 might be more stronger if the model is compared with that, right now there are no other baselines to compare performance with.

2) It might help to mention how long it takes to find the bestLDS initialization, if we are going to claim that it help faster convergence. It would also be nice to see total time to convergence, including computing the initialization using bestLDS.

3) I'm not really clear what the take-away for the real-world experiment is: the last paragraph before the discussion is very dense and hard to follow. Here too, it might be worth comparing with the Poisson LDS approach of Buesing et al.

---

> ### Author Response · Authors · 2023-06-09
> **Response to Reviewer 1 (Part 1)**
>
> We thank the reviewer for their insightful comments and suggestions. We agree that addressing many of these points will strengthen the manuscript. Given that the same topics often arise at different points in the review, for conciseness and clarity we have organized our responses below according to topic, along with references to the places in which they appear in the review. We have also uploaded a revised version of the manuscript with the changes outlined below.
>
> **Novelty (Negatives 1, Requested Change 1)**
>
> > As far as I can tell, this paper is a fairly straightforward extension of Buesing et al from the Poisson case to the Bernoulli case. The authors here use the same principle: the observations Y are a transformation of Gaussian intermediate outputs Z, and learning the parameters of the system involves inferring the mean-covariance structure of Z from Y. The main technical problem here is that for Bernoulli variables, since the mean equals the mean square, the system of equations is underdetermined. The authors solution is just to set the diagonal of the covariance of Z to be one and solve for the remaining covariance variables. I'm not convinced this is a significant methodological innovation. The paper also allows inputs u to the linear-dynamical system, but dealing with this is also a fairly immediate application of Buesing et al.
>
>
> > I think the paper would benefit with a clearer discussion of the technical novelty, in its current form it does not seem like a significant advancement over Buesing et al. The authors might also consider comparing with this approach.
>
> On the point of technical novelty, we would like to note that TMLR has stated that it will evaluate submissions based on whether “the claims made in the submission [are] supported by accurate, convincing, and clear evidence” and that “at least some individuals in TMLR’s audience [would] be interested in knowing the findings” of the paper. If papers meet this criteria, they should be considered for acceptance even if “the contribution or significance of the work is modest.”
>
> That being said, our paper IS technically novel in that it introduces a new technical contribution to the field of spectral learning (namely the moment equations outlined in section 3.3*) for a distribution class that is widely relevant in machine learning research while also providing a close characterization of the input-related parameters, which had not been performed previously. Note this methodological innovation requires more than simply setting the diagonal of the covariance of Z to one, as the reviewer suggests. That is but one step taken as part of the broader contribution, which includes figuring out the entire derivation of \Sigma. Further, we detail the innovations and the ways in which these innovations are relevant to TMLR’s audience in the last two paragraphs of the introduction.
>
> We have also added a section comparing with the Buesing et al. approach (see “Comparison to Poisson LDS”).
>
> **Probit transformations (Negatives 2, Requested Change 4)**
>
> > The proposed method only works for probit transformations, where the sysyem of nonlinear equations relating the moments of Y and Z can be written explicitly.
>
>
> > The method also only applied for probit-transformations, which is not mentioned clearly upfront (in the abstract/introduction).
>
> We acknowledge that the proposed method only works for probit transformations and have edited the text accordingly to be more explicit about this.
>
> **Evaluation metrics (Negatives 3)**
>
> > Evaluation metrics seem ad hoc and not well motivated. Why is recovery of A measured using eigenvalues, and C using the angle between the subspaces? I don't think the paper clearly enough discusses the effect on estimating C of fixing the diagonal of the covariance of z.
>
> An issue in assessing the accuracy of the recovered parameters for LDS models is that they are not uniquely identifiable. In general, SSID algorithms can only recover these parameters up to a similarity transformation. We go into detail on these transformations for individual system parameters in section 4.1. The evaluation metrics we chose are invariant under similarity transformations.
> We agree that it is worthwhile to examine the effect on estimating C when the diagonal of the covariance of z is not 1. We have now added an analysis that examines this point in Supplemental Figure A.1. The reviewer is right to surmise that there is an effect on C, although the recovery of the other parameters is less affected. Note that despite this effect, the many evaluations we perform in the paper demonstrate that bestLDS nonetheless provides a good estimate of the system in many contexts and is a useful initializer of EM.

---

> > ### Author Response · Authors · 2023-06-09
> > **Response to Reviewer 1 (Part 2)**
> >
> > **Comparison to Poisson LDS (Question 1, Question 3, Requested Change 1)**
> >
> > > Can we directly apply the estimator for the Poisson observations from Buesing et al, treating the Bernoulli outputs as outputs from a Poisson distribution with small rate parameter? Table 1 might be more stronger if the model is compared with that, right now there are no other baselines to compare performance with.
> >
> >
> > > I'm not really clear what the take-away for the real-world experiment is: the last paragraph before the discussion is very dense and hard to follow. Here too, it might be worth comparing with the Poisson LDS approach of Buesing et al.
> >
> > > I think the paper would benefit with a clearer discussion of the technical novelty, in its current form it does not seem like a significant advancement over Buesing et al. The authors might also consider comparing with this approach.
> >
> > We thank the reviewer for this comment. We agree that comparing against the Buesing et al. pLDS approach would strengthen the paper, and we are currently running simulations to do this.  In readdressing this analysis, it was also brought to our attention that while the simulated data is generated from a probit-Bernoulli model, the inference code used to compute the log-evidences assumed a logit model. To address this inconsistency within the review time period, we have adjusted the inference code to account for a scale factor on the observation parameters C and D to align the probit and logit models, an adjustment which will only further improve our bestLDS results. As such, we are currently in the process of updating the results for all three datasets and model comparisons.  Our findings thus far indicate that bestLDS greatly outperforms pLDS. For example, the average test log-evidences for dataset C (50k / 256k) are as follows:
> >
> > true: -41161 / -211821
> >
> > bestLDS: -40269 / -207713
> >
> > pLDS: -42799 / -226886
> >
> > If the reviewers are willing to give us a few more days before submitting final comments, we will upload another revised manuscript as soon as possible with all the new log-evidence values in Table 1, including a new row with the log-evidence values for pLDS, once all model comparisons are complete (with the exception of pLDS results for dataset A, as the pLDS code does not allow for analysis of data where q < p. Note that this points to an additional strength of our method).
> >
> > **bestLDS computation time (Question 2, Requested Change 3)**
> >
> > > It might help to mention how long it takes to find the bestLDS initialization, if we are going to claim that it help faster convergence. It would also be nice to see total time to convergence, including computing the initialization using bestLDS.
> >
> >
> > > I would like to see some theoretical and empirical discussion of the cost of solving the nonlinear moment conditions.
> >
> > We agree that including a more detailed quantitative assessment of the time it takes to find the bestLDS initialization could be useful. As such, we have now added two analyses to the paper. First, we have added a supplemental figure to the manuscript characterizing the time cost as a function of dataset size for two datasets with different dimensionality (see new Supplemental Figure A.2). We have also incorporated the amount of time it takes to run the bestLDS estimator for each dataset into the EM convergence times table (see updated Table 2).
> >
> > We find that the time it takes to do the moment conversions does not override the substantial cost savings that are gained through the subsequent EM fitting process. In the slowest case that we evaluated (dataset D, q=5, N=256k), the time to run the estimator is 8.7 seconds, which is inconsequential relative to the time to run EM to convergence (34.67 minutes). The results are also favorable when examining a smaller dataset in which one could reasonably expect EM to converge more quickly (dataset D, q=5, N=50k). In this case, the time to run the estimator is 5.12 seconds and the time for EM to converge is 8.4 minutes.
> >
> > On the theoretical front, we note in the discussion that the total number of unknown variables in the system of equations scales as O((kq)^2). As such, increasing either the Hankel parameter k or the system output dimensionality q can strongly affect the time taken for moment conversion. These points are demonstrated indirectly in the new Supplemental Figure A.2 as well, by showing the estimator run times for two datasets with different dimensionalities in q, p, and k.

---

> > > ### Author Response · Authors · 2023-06-09
> > > **Response to Reviewer 1 (Part 3)**
> > >
> > > **Description of real data results (Question 3)**
> > >
> > > > I'm not really clear what the take-away for the real-world experiment is: the last paragraph before the discussion is very dense and hard to follow. Here too, it might be worth comparing with the Poisson LDS approach of Buesing et al.
> > >
> > > We agree that the referenced paragraph is dense and have clarified the take-aways for the real-world data. While we agree that comparing with pLDS on the mouse data would be a worthwhile comparison, unfortunately the code published by Buesing et al. does not allow for data in which the dimensionality of the observation is lower than the dimensionality of the latent (q < p), and in this dataset we have q = 1 and p = 3. This is indeed another advantage of our approach, as it functions well in this regime and thus can be used on neural data where pLDS cannot.
> > >
> > > **Moment conversion equations (Requested Change 2)**
> > >
> > > > I think the paper would benefit from a slightly clearer discussion of how the nonlinear systems of equations are solved. RIght now, it seems like this is a high-dimensional problem, but if I understand correctly, this is just a collection of uncoupled low-dimensional equations. It would also be nice if the paper also dwelt a little more on the situation when the equations don't have a solution, this is mentioned only very perfunctorily.
> > >
> > > To clarify how the nonlinear system of equations is solved, we have added a note in our summary of the steps for the bestLDS estimator at the end of Section 3.3 to be more explicit about how \Sigma^{zz} and \Sigma^{uz} are related. That is, we want to stress that in terms of the coupling, when solving for the elements of \Sigma^{zz}, we have a constraint for every off-diagonal term (eq. 5). Furthermore, when solving for \Sigma^{uz}, there is a constraint for every element in the block, which is coupled with \Sigma^{zz} (see eq. 6).
> > > We agree that a more in-depth discussion of the moment conditions and where they don’t have a solution is warranted, and we have clarified our comments on this in the main text in section 3.3 – though we would like to stress that in the many simulations we’ve run, as well as in our analysis of real world data, this has never occurred. In particular, we note that we expect such regimes (where the vast majority of the data points are either 0s or 1s) are rare in real-world applications and so did not find the need to rely on additional steps to deal with such high autocorrelation cases in our analyses.
> > >
> > > **Title (Requested Change 4)**
> > >
> > > > I think "for decision-making" in the title is unnecessary and a bit confusing.
> > >
> > > Thank you for pointing this out. We have removed “for decision-making” in the title and have clarified in the abstract that the method assumes a probit transformation.

---

### Review · Reviewer_Ex51 · 2023-05-12

**Summary Of Contributions:**

This work proposes an estimation method for a linear Gaussian state space model, where the continuous observation in the typical formulation is treated as latent, and the observation is instead given as a Bernoulli variable depending on the latent.
A technical contribution of this work is to extend the N4SID method, a technique for continuous models, to this binary setting.

Specifically, the N4SID method requires the QR decomposition of the (unavailable) input-latent Hankel matrix. Authors proposed to use the Cholesky decomposition of the covariance of the input-latent Hankel matrix as the R matrix.
Major findings of the paper are the following: in a well-specified setting, the method (empirically) seems to yield a consistent estimate; the authors also show that this serves as a good initialisation for the EM algorithm.


**Audience:**

Yes

**Broader Impact Concerns:**

The work is methodological, and I do not see any potential ethical implication.


**Claims And Evidence:**

Yes

**Requested Changes:**

Please address the weakness comment above.

**Strengths And Weaknesses:**

## Strengths
* The paper is generally well-written.
* The paper deals with a common setting in sequential data analysis, and some TMLR readers would appreciate this work.
* Extensive experiments are performed to support the proposed method, including an experiment with a real-world behavioural dataset.

## Weakness
* The evaluation of the method could be made stronger. I would have preferred to see more synthetic experiments to investigate failure modes of the method.
  * The synthetic data experiment seems to deal with a well-specified setting only. The method assumes the unit covariance diagonal of the marginal of the latent. What happens if this assumption is violated?
  * For synthetic data, we can generate $z$. It might be helpful to compare against the exact N4SID to see the effect of the moment conversion.

---

> ### Author Response · Authors · 2023-06-09
> **Response to Reviewer 2**
>
> We thank the reviewer for their useful comments and insights. We agree that these points are particularly important to investigate and will add significant value to the manuscript. For conciseness and clarity we have organized our responses by the topics as presented by the reviewer. We have also uploaded a revised version of the manuscript with the changes outlined below.
>
> **Failure mode: non-unit diagonal covariance (Weakness 1)**
>
> > The synthetic data experiment seems to deal with a well-specified setting only. The method assumes the unit covariance diagonal of the marginal of the latent. What happens if this assumption is violated?
>
> We agree with the reviewer that investigating how the method performs when there is non-unit diagonal covariance is warranted. Outside the well-specified setting, the estimator is naturally less accurate. We replicated the error metrics analysis from Figure 2 without forcing the diagonal of the covariance matrix to be 1 and found that this has different effects by parameter (see new Supplemental Figure A.1). This is most notable in the error metric for C, which plateaus around 0.5 – but the other parameters are largely unaffected. Inference of the dynamics matrix A remains consistent and error shrinks to 0 with increased N, and the error metrics for D and the gain matrix decrease with N. Furthermore, we want to emphasize that the performance of the estimator on the real data suggests that even (far) outside the well-specified setting, bestLDS is highly useful.
>
> **Failure mode: comparison to N4SID directly on z (Weakness 2)**
>
> > For synthetic data, we can generate  z. It might be helpful to compare against the exact N4SID to see the effect of the moment conversion.
>
> We thank the reviewer for their suggestion and agree that comparing the results to running subspace identification directly on z would be useful. As part of the development process, we indeed had tested the difference between applying exact N4SID directly to z and moment-conversion + N4SID to y. As one would expect, the exact inference is better, especially in regimes that violate the statistical assumptions of our estimator. As shown in the many other evaluations we performed, this does not preclude the utility of the method in many settings, whether as a replacement for exact inference or as an initialization for EM (which ameliorates the issues created by the statistical assumptions of our estimator). Nonetheless, for completeness we have included a plot of the difference between the exact and estimated inference in the supplement (see new Supplemental Figure A.3).

---

### Review · Reviewer_1ZBy · 2023-05-26

**Summary Of Contributions:**

The authors propose a spectral learning method for latent linear dynamical systems (LDS) with Bernoulli observations. The proposed method operates on transformations of the first and second moments of the time series. The proposal is validated experimentally.



**Audience:**

No

**Claims And Evidence:**

No

**Requested Changes:**

Perhaps this article is out of the scope of TMLR, I would have expected a more general model formulation, a more principled methodology to analyse the model and a thorough experiment tal validation.

**Strengths And Weaknesses:**

The article is well written and the literature revision is thorough.

Despite the fact that the article is clearly presented, I had a tough time understanding the novel contribution of the manuscript. It is not entirely clear when the authors describe their setup and when they describe their proposal.

The authors proposed a model for time series---eq. (1)---together with a technique to analyse that model. Then, this proposal is tested on synthetic and real-world data. In this context, I can't identify a contribution to the machine learning community from the proposal of the authors. The fixed model together with the mechanism to analyse it do not constitute a novelty within ML.

The experimental validation is also far from convincing. The proposal is tested on synthetic data (Secs 4.1 and 4.2), where best LDS is compared against the ground truth or initialisation heuristics, but not other benchmarks. It is not clear to me if the known parameters used to produce realisations of the models are used in this experiments.

In the last experiment, though real-world data were considered, there are also no comparison: all claims are based on inspection of the results of best LDS rather than on a quantitative comparison against other methods. As a non-expert on what to expect from the mice experiment, the feeling I have after reading about the experiment is inconclusive: the method was applied and we obtained results, however, I wouldn't know how to assess them.

---

> ### Author Response · Authors · 2023-06-09
> **Response to Reviewer 3**
>
> We thank the reviewer for the concerns they have raised. We have clarified our position on these points below and hope that our responses serve to ameliorate the reviewer’s concerns about the technical innovations and relevance of our manuscript for the TMLR audience. We have also uploaded a revised version of the manuscript with the changes outlined below and in response to the other reviews.
>
> **Technical novelty (Weakness 1)**
>
> > The article is well written and the literature revision is thorough. Despite the fact that the article is clearly presented, I had a tough time understanding the novel contribution of the manuscript. It is not entirely clear when the authors describe their setup and when they describe their proposal.
>
>
> > The authors proposed a model for time series---eq. (1)---together with a technique to analyse that model. Then, this proposal is tested on synthetic and real-world data. In this context, I can't identify a contribution to the machine learning community from the proposal of the authors. The fixed model together with the mechanism to analyse it do not constitute a novelty within ML.
>
> We thank the reviewer for highlighting this concern. To be clear, our contribution is not the fundamental construction of a linear dynamical system (eq. 1). Indeed, linear dynamical systems are already in use throughout the neuroscience and ML communities (see Background and Related Work section). Rather, our contribution is the derivation of a spectral estimator for that model that returns consistent and accurate estimates of the model parameters (i.e. eqs. 5 and 6) such that it can be used in place of or in conjunction with more traditional inference methods, such as the Expectation Maximization algorithm, to significantly improve computation time. This is of great importance to the ML community, especially as high-dimensional and large-scale datasets become commonplace, in order to fit such models in a timely, cost-effective, and energy-efficient manner. From a theory perspective, this involved deriving the moment conversion conditions outlined in section 3.3. We then empirically analyzed the properties of this estimator, including in regimes previously untested by the literature.
>
> **Experimental validation (Weakness 2)**
>
> > The experimental validation is also far from convincing. The proposal is tested on synthetic data (Secs 4.1 and 4.2), where best LDS is compared against the ground truth or initialisation heuristics, but not other benchmarks. It is not clear to me if the known parameters used to produce realisations of the models are used in this experiments.
>
> We thank the reviewer for this comment. We are working on adding more comparisons than are already present in the paper (i.e. the comparison to the pLDS model by Buesing et al. (2012) in Table 1, as suggested by Reviewer 1) to ameliorate this issue. We need a few more days to finish this analysis, at which point we will upload another revised manuscript with the new log-evidence values in Table 1). However, we do want to note that there is a comparison to a similar method (Gaussian LDS) in Figure 3 and that comparisons to ground truth and other initialization methods are in fact a useful way to assess the performance of our estimator in this context.
>
> **Real data (Weakness 3)**
>
> > In the last experiment, though real-world data were considered, there are also no comparison: all claims are based on inspection of the results of best LDS rather than on a quantitative comparison against other methods. As a non-expert on what to expect from the mice experiment, the feeling I have after reading about the experiment is inconclusive: the method was applied and we obtained results, however, I wouldn't know how to assess them.
>
> Respectfully, we disagree with the reviewer’s assessment of the real-data analysis. In Figure 4e, we compare bestLDS and bestLDS + the EM algorithm against three other baseline models and show that bestLDS + EM outperformed them all. We also have a supplemental figure showing the comparison between the recovered parameters and the impulse response plots for bestLDS and bestLDS+EM (Supplemental Figure A.4). Furthermore, in the text, we highlight that the insights gained by inspecting the learned parameters and the impulse responses correspond with previous analysis performed on the same dataset with a different statistical model.
>
> **Scope (Requested Change 1)**
>
> > Perhaps this article is out of the scope of TMLR, I would have expected a more general model formulation, a more principled methodology to analyse the model and a thorough experiment tal validation.
>
> We thank the reviewer for the concerns they have highlighted. We respectfully disagree that the work (a largely methodological paper presenting an estimation method for an ML model) is out of scope, and we hope that our clarifying comments above as well as the changes we are implementing in response to other reviewer comments will ameliorate the reviewer’s concerns.

---

### Decision · Action_Editors · 2023-07-10

**Recommendation:** Accept with minor revision

**Comment:**

The paper proposes spectral estimation for linear Gaussian state space models with binary observables.  It can be used as low-computation (and somehow interpretable) estimator, as well as initialization of the EM algorithm.  Experiments show good performance.

Reviewers originally mentioned the following pros and cons:

Pros:
- Well written
- Relevant problem setting
- Experiments support the proposed method.

Cons:
- Limited novelty (straight-forward extension of Buesing et al.)
- Weak experiments (lack of baselines in some experiments, miss-specified case)
- Evaluation metrics seem ad hoc.

The authors revised the manuscript, addressing most of the concerns.  The novelty is at least clear.  Although it's not innovation, it matches the TMLR criterion.
In the revision, baseline methods are added, and the miss-specified cases are discussed.  The authors' arguments on the evaluation metrics are convincing.

Overall, the paper has been significantly improved after revision, and the revised paper satisfies the TMLR criteria for acceptance.

I have a question, which is the reason why the decision is conditional.  Please make this point clear with minor revision:
In Table 1, bestLDS indeed gives the highest evidence.  My question is why it is even better than "true".  I can't find clear explanation about what "true" means, and I'd assume that it is a kind of oracle, which gives an upper-bounds of performance of practical methods.  If this would be the case, the reason why bestLDS gives higher evidence should be discussed.
If not and "true" is just another baseline method, which bestLDS outperforms, then the method "true" should be clearly explained, and what makes the "true" method weak even though it uses the true parameters should be discussed.

**Audience:**

yes

**Claims And Evidence:**

yes